



# Using Network Theory and Machine Learning to predict El Niño

Peter D. Nooteboom[1,3], Qing Yi Feng[1,3], Cristóbal López[2], Emilio Hernández-García[2], and Henk A. Dijkstra[1,3]

[1]Institute for Marine and Atmospheric Research Utrecht (IMAU), Department of Physics, Utrecht University, The Netherlands
[2]Instituto de Física Interdisciplinar y Sistemas Complejos (IFISC, CSIC-UIB), University of the Balearic Islands, Spain
[3]Centre for Complex Systems Studies, Utrecht University, The Netherlands

*Correspondence to:* Peter Nooteboom (p.d.nooteboom@uu.nl)

**Abstract.** The skill of current predictions of the warm phase of the El Niño Southern Oscillation (ENSO) reduces significantly beyond a lag of six months. In this paper, we aim to increase this prediction skill at lags up to one year. The new method to do so combines a classical Autoregressive Integrated Moving Average technique with a modern machine learning approach (through an Artificial Neural Network). The attributes in such a neural network are derived from topological properties of Climate

Networks and are tested on both a Zebiak–Cane-type model and observations. For predictions up to six months ahead, the results of the hybrid model give a better skill than the CFSv2 ensemble prediction by the National Centers for Environmental Prediction (NCEP). Moreover, results for a twelve-month lead time prediction have a similar skill as the shorter lead time predictions.

## 1 Introduction

Approximately every four years, the sea surface temperature (SST) is higher than average in the eastern equatorial Pacific (Philander, 1990). This phenomenon is called an El Niño and is caused by a large-scale ocean-atmosphere interaction between the equatorial Pacific and the global atmosphere (Bjerknes, 1969), referred to as El Niño/Southern Oscillation (ENSO). It is the dominant mode of variability at interannual time scales and has teleconnections worldwide. As El Niño events cause enormous damage worldwide, skillful predictions, preferable for lead times up to one year, are highly desired.

So far, both statistical and dynamical models are used to predict ENSO (Chen et al., 2004; Yeh et al., 2009; Fedorov et al., 2003). However, El Niño events are not predicted well enough up to six months ahead due to the existence of the so-called predictability barrier (Goddard et al., 2001). Some theories indicate this is due to the chaotic, yet deterministic, behavior of the coupled atmosphere-ocean system (Jin et al., 1994; Tziperman et al., 1994). Others point out the importance of atmospheric noise, acting as a high frequency forcing sustaining a damped oscillation (Moore and Kleeman, 1999).

Recently, attempts have been made to improve the ENSO prediction skill beyond this spring-predictability boundary, for example by using machine learning (ML) (Wu et al., 2006) methods, also combined with network techniques (Feng et al., 2016). ML has shown to be a promising tool in other branches of physics, outperforming conventional methods (Hush, 2017). In addition, ML did a better job than humans in a game of GO, which is difficult for Artificial Intelligence (AI) since it requires



intuition and creative thinking (Silver et al., 2016). As the amount of data in the climate sciences is increasing, ML methods such as Artificial Neural Networks (ANN), are becoming more interesting to apply to prediction studies.

Briefly, ANN is a system of linked neurons that describes, after optimization, a function from one or more input variables to the output variable(s). Generally, two different approaches can be considered when applying the ANN. The first is to use a

complicated ANN structure with a lot of layers in the network and many input variables (or attributes). This approach is situated on the deep learning part of the ML spectrum and is believed to filter the important information from the attributes itself. The deep learning approach requires a lot of input data and is computationally intensive. Therefore, simpler ANN structures are used in this article. However, techniques will have to be applied in order to reduce the amount of input variables and select the important ones, to make the problem appropriate for the simpler ANN. This reduction and selection problem can be tackled in

many ways, which are crucial for the prediction. The main issue in these methods is, however, what attributes to use for ENSO prediction.

Complex networks turn out to be an efficient way to represent spatio-temporal information in climate systems (Tsonis et al., 2006; Steinhaeuser et al., 2012; Fountalis et al., 2015) and can be used as an attribute reduction technique. These Climate Networks (CN) are in general constructed by linking spatio-temporal locations that are significantly correlated with each

other according to some measure. It has been demonstrated that relationships exist between topological properties of CNs and nontrivial properties of the underlying dynamical system (Tupikina et al., 2014; Deza et al., 2014; Stolbova et al., 2014), also specifically for ENSO (Gozolchiani et al., 2011, 2008; Wang et al., 2015). CNs already appeared to be a useful tool for more qualitative ENSO prediction, by considering a warning of the onset of El Niño when a certain network property exceeds some critical value (Ludescher et al., 2014; Meng et al., 2016; Rodríguez-Méndez et al., 2016).

In this paper, a hybrid model is introduced for ENSO prediction. The model combines the classical linear statistical method of Autoregressive Integrated Moving Average (ARIMA) and an ANN method. ANN is applied to predict the residual, due to the nonlinear processes, that is left after the ARIMA forecast (Wu et al., 2006). To motivate our choice for attributes in the ANN, we use an intermediate complexity model which can adequately simulate ENSO behavior, the Zebiak-Cane (ZC) model (Zebiak and Cane, 1987). Network variables are chosen as attributes, such that they are related to another physical quantity

capturing information about the system, but spatial information is conserved.

Section 2 briefly describes the ZC model, the methods considering both the CNs and ML and the used data from observations. In Sect. 3, the network methods are first applied to the ZC model. Second, the attributes selected for observations are presented. These attributes, among which there is a network variable, are applied in the hybrid prediction model in Sect. 4, which discusses the skill of this model to predict El Niño. The paper concludes with a summary and discussion in Sect. 5.

## 2   Observational data, models and methods

### 2.1   Data from observations

As observational data, we use the sea surface height (SSH) from the weekly ORAP5.0 (Ocean ReAnalysis Pilot 5.0) reanalysed dataset of ECMWF from 1979 to 2014 between $140°E$ to $280°E$ and $20°S$ to $20°N$.



For recent predictions, the SSALTO/DUACS altimeter products are used for the same spatial domain, since the SSH is available from 1993 up to present in this dataset. The SSALTO/DUACS altimeter products were produced and distributed by the Copernicus Marine and Environment Monitoring Service (CMEMS) (http://www.marine.copernicus.eu).

In addition, the HadISST dataset of the Hadley center has been used for the SST and the NCEP/NCAR Reanalysis dataset
for the wind stress from 1980 to present (Rayner et al., 2003).

To quantify ENSO, the NINO3.4 index is used, i.e., the three-month running mean of the average SST anomaly in the extended reconstructed sea surface temperature (ERSST) dataset between $170°$W to $120°$W and $5°$S and $5°$N (Huang et al., 2015).

The warm water volume (WWV), being the integrated volume above the $20°C$ isotherm between $5°N$-$5°S$ and $120°E$-
$280°E$, is determined from the temperature analyses of the Bureau National Operations Centre (BNOC) (https://www.pmel.noaa.gov/elnino/ ocean-heat-content-and-enso).

## 2.2 The Zebiak-Cane model

The ZC model (Zebiak and Cane, 1987) represents the coupled ocean-atmosphere system on an equatorial $\beta$-plane in the equatorial Pacific (Fig. 1). We use the numerically implicit version of this model (van der Vaart et al., 2000; Von Der Heydt et al., 2011) as in Feng (2015). In the ZC model, a shallow-water ocean component is coupled to a steady shallow-water Gill

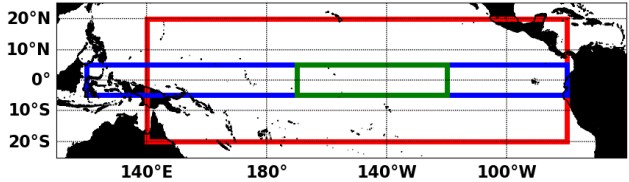

**Figure 1.** Pacific area (red rectangle) from $140 - 280°$E and $-20 - 20°$N , the NINO3.4 area (green rectangle) from $170 - 120°$W and $-5°$S$-5°$N and the WWV area (blue rectangle) from $120 - 280°$E and $-5°$S$-5°$N.

(Gill, 1980) atmosphere model. The atmosphere is driven by heat fluxes from the ocean, depending linearly on the anomaly of the sea surface temperature $T$ with respect to a radiative equilibrium temperature $T_0$. The zonal wind stress $\tau^x$ is the sum of a coupled and an external part:

$$\tau^x = \tau^x_{ext} + \tau^x_c. \tag{1}$$

The external part $\tau^x_{ext}$ is independent of the coupling between the atmosphere and ocean. It represents a weak easterly wind stress due to the Hadley circulation. It is assumed to be zonally constant and depends on latitude according to:

$$\tau^x_{ext} = -\tau_0 e^{-\frac{1}{2}\left(\frac{y}{L_a}\right)}. \tag{2}$$





Here $\tau_0 \sim 0.01\ Pa$, $L_a$ the atmospheric Rossby deformation radius and $y$ is the meridional coordinate. The coupled part of the zonal wind stress $\tau_c^x$ is proportional to the zonal wind from the atmospheric model; the meridional component of the wind stress is neglected in this model.

As shown in van der Vaart et al. (2000), the parameter measuring the magnitude of the ocean-atmosphere coupled processes
is the coupling strength $\mu$. Without any included noise, a temperature anomaly damps out to a constant value and a stationary state if $\mu < \mu_c$, where $\mu_c$ indicates a critical value. However, if the coupling strength exceeds the critical value $\mu_c$, a supercritical Hopf bifurcation occurs. A perturbations then does not decay, but an oscillation is sustained with a period of approximately four years.

Three positive feedbacks related to the thermocline depth, upwelling and zonal advection, can cause the amplification of
SST anomalies (Dijkstra, 2006) while the oscillatory behavior associated with ENSO is caused by negative delayed feedbacks. The 'classical delayed oscillator' paradigm assumes this negative feedback is caused by waves through geostrophic adjustment, controlling the thermocline depth. A complementary, different view is the 'recharge/discharge oscillator' (Jin, 1997), also regarding oceanic waves excited through oceanic adjustment. The waves excited to preserve the Sverdrup balance are responsible for a transport of warm surface water to higher latitudes, discharging the warm water in the tropical Pacific. The thermocline
depth is raised, resulting in more cooling of SST. The warm water volume (WWV) is the variable generally used to capture how much the tropical Pacific is 'charged.'

Apart from the coupled ocean-atmosphere processes, ENSO is also affected by fast processes in the atmosphere, which are considered as noise in the ZC model. An important example of atmospheric noise are the so-called westerly wind bursts (WWB). These are related to the Madden-Julian oscillation (Madden and Julian, 1994). The WWB is a strong westerly anomaly
in the zonal wind field, occurring every forty to fifty days and lasting approximately a week. The effect of the noise on the model behavior depends on whether the model is in the super- or sub-critical regime (i.e whether $\mu$ above or below $\mu_c$). If $\mu < \mu_c$, the noise excites the ENSO mode, causing irregular oscillations. In the supercritical regime, a cycle of approximately four years is present, and noise causes a larger amplitude of ENSO variability.

To represent the atmospheric noise in the model, we obtained a residual of the wind stress from observations. We used the
ERSST over the Pacific for the period 1978-2004 and the Florida State University pseudo-wind-stress data (Legler and Brien, 1988) for the same period. First the part of the wind stress anomalies linearly related to the SST anomalies are subtracted from the wind stress anomalies. Then the residual is projected on its Empirical Orthogonal Functions (EOFs). The six EOFs that explain most variability were considered to describe the spatial patterns of the noise and their Principal Components (PC) were used to construct the time series (Feng and Dijkstra, 2016). These time series are modeled as an independent first order Auto
Regressive (AR(1)) process by:

$$x_{t+1} = ax_t + \sigma\epsilon_t, \tag{3}$$

where $a$ is the lag-1 autocorrelation of each PC and $\sigma\epsilon_t$ is the white noise with variance $\sigma$. Since weekly data are considered, every discrete time step is one week.





## 2.3 Network variables

Different methods can be used to construct CNs. Here only the Pearson correlation (appendix A1) will be considered for the construction of two different types of CN. In the first method an undirected and unweighted network is constructed. Network nodes are model or observation grid positions $i$ and the links are stored in a symmetric adjacency matrix $A$, where $A_{ij} = 1$ if node $i$ is connected to node $j$ and $A_{ij} = 0$ otherwise. $A_{ij}$ is defined by:

$$A_{ij} = \Theta\left(|R_{ij}| - \epsilon\right) - \delta_{ij}. \tag{4}$$

Here $R_{ij}$ is the Pearson correlation between node $i$ and $j$, $\epsilon$ is the threshold value and $\Theta$ denotes the Heaviside function. Hence, if the Pearson correlation exceeds the threshold $\epsilon$, the two nodes will be linked. The $\delta_{ij}$ is the Kronecker delta function, implemented to prevent connection of nodes with themselves.

The second method creates a weighted, undirected network and will be applied for only one network variable later. The cross-correlation $C_{ij}(\Delta t)$ at lag $\Delta t$, i.e. the Pearson correlation between the variables $p_i(t)$ and $p_j(t + \Delta t)$ is considered. Then the weights between the nodes are calculated by:

$$W_{ij} = \frac{\max_{\Delta t}(C_{ij}) - \text{mean}(C_{ij})}{\text{std}(C_{ij})}. \tag{5}$$

Here $\max_{\Delta t}$ denotes the maximum, std the standard deviation and mean the mean value over all time steps that are considered.

From the CNs, we construct several properties. From the unweighted network we compute the local degree $d_i$ of node $i$ in the CN as,

$$d_i = \sum_j A_{ij}, \tag{6}$$

i.e. degree is equal to the amount of nodes that are connected to node $i$. The spatial symmetry of the degree distribution is of interest, since it informs where most links of the network are located. More specifically, our interest will be in the symmetry in the zonal direction in a network. Therefore, the skewness of the meridional mean of the degree in the network is calculated. This defines the zonal skewness of the degree distribution in a network.

Second, percolation theory is considered, describing the connectivity of different clusters in a network. It has been found that the connectivity of some CN increases when approaching an El Niño and decreases afterwards (Rodríguez-Méndez et al., 2016), as local correlations between points increase and decrease. At such a percolation-like transition, the addition of only a few links can cause a considerable part of the network to become connected. Before the percolation transition, clusters of small sizes will form. Therefore the variable $c_s$ will warn for the transition:

$$c_s = \frac{s n_s}{N}. \tag{7}$$

Here $n_s$ is the amount of clusters of size $s$ and $N$ the size (i.e. the total amount of nodes) of the network. Thus $c_s$ is the fraction of nodes that are part of a cluster of (generally small) size $s$. For another variable related to the percolation-like transitions, links are added to a network one by one, adding the link with the largest weight first (Eq. (5)). At every step $T$ that a link is





added, the size of the largest cluster $S_1(T)$ is calculated. At the point of the percolation transition, $S_1(T)$ increases rapidly. The size of this jump is $\Delta$:

$$\Delta = \max\left[S_1(2) - S_1(1),\ldots,S_1(T+1) - S_1(T),\ldots\right]. \tag{8}$$

The quantity $\Delta$ can be used to capture the percolation-like transition (Meng et al., 2016).

The final two CN properties are derived from a so-called NetOfNet approach. This is a network constructed with the same methods as previously, but using multiple variables at each grid point (as specified later in the results). This gives a network consisting of the networks from the different variables interacting with each other. Only NetOfNet of two different variables are considered. First, the cross clustering contains information about the interaction between two unweighted networks. The local cross clustering of a node is the probability that two connected nodes in the other network are also connected to each

other. The global cross clustering $C_{vw}$ is the average over all nodes in subnetwork $G_v$ of the cross clustering between $G_v$ and $G_w$:

$$C_{vw} = \frac{1}{N_v}\sum_r \frac{1}{k_r(k_r - 1)}\sum_{p \neq q} A_{rp}A_{pq}A_{qr}. \tag{9}$$

Here $r$ is a node in subnetwork $G_v$ of size $N_v$, $p$ and $q$ are the nodes in the other subnetwork $G_w$ and $k_r$ denotes the cross degree of node $r$ (i.e. amount of cross links node $r$ has with the other subnetwork). In addition to the clustering coefficient,

also the algebraic connectivity ($\lambda_2$) is considered (appendix A2) within a NetOfNet. This variable includes the diffusion of information within one or multiple networks.

## 2.4    Hybrid prediction model

A hybrid model (Valenzuela et al., 2008) will be applied to predict ENSO, in which the observation $Z_t$ at time $t$ is represented by

$$Z_t = Y_t + N_t. \tag{10}$$

Here $Y_t$ is modelled by a linear process and $N_t$ by a ML type technique. Let $\tilde{Y}_t$ be the prediction of the part $Y_t$ using ARIMA, then $Z_t - \tilde{Y}_t$ is the residual with respect to the observed value. This residual will be predicted by the feedforward ANN:

$$\tilde{N}_t = f\left(x_1(t),\cdots,x_N(t)\right). \tag{11}$$

Here $f$ is a nonlinear function of the $N$ attributes $x_1(t),\cdots,x_n(t)$ and $\tilde{N}_t$ the prediction of residual $Z_t - \tilde{Y}_t$ at time $t$. Notice

the nonlinear function $f$ does not depend on history, whereas the ARIMA part $\tilde{Y}$ does. The final prediction $\tilde{Z}_t$ of the hybrid model:

$$\tilde{Z}_t = \tilde{Y}_t + \tilde{N}_t. \tag{12}$$

Previous work showed the results of a hybrid model are in general more stable and reduce the risk of a bad prediction, compared to a single prediction method (Hibon and Evgeniou, 2005).



This scheme describes a 'supervised' model, implying that the predictant is 'known.' This known quantity is the NINO3.4 index. The standard procedure for supervised learning is to optimize the ML method on a 'training set' to define an optimal model, which predicts ENSO with a certain time ahead. This function will then be tested on a test set. Here a training set of 80 % and a test set of 20 % of the total time series is used. The data set can be represented by a $T \times N$ matrix, where $T$
represents the length of the time series and each time $t = 1,...,T$ has a set of $N$ attributes $x_1(t),...,x_N(t)$. Note that, since we are predicting time series, for any training set $[t_i^{train}, t_f^{train}]$ and test set $[t_i^{test}, t_f^{test}]$, $t_i^{test} > t_f^{train}$ must hold (where $t_i^{train}, t_f^{train}, t_i^{test}, t_f^{test} \in [1, T]$). In the following, we describe more in detail the different parts of this hybrid prediction method.

First, the training set is used to optimize an ARIMA($p,d,q$) process for the NINO3.4 time series. The standard method
maximizing the log likelihood function is used to fit $\alpha_1, \cdots, \alpha_p, \beta_1, \cdots, \beta_q$, such that $\sum_t \varepsilon_t^2$ is minimized for time series $Z_t$ with $t$ in months:

$$Z_t = \sum_{i=1}^{d} Z_{t-i} + \sum_{j=1}^{p} +\alpha_j Z_{t-j} + \sum_{k=1}^{q} \beta_k \varepsilon_{t-k} + \varepsilon_t, \tag{13}$$

where $\varepsilon_t$ is the residual, differencing order $d$ determines the amount of differencing terms, $p$ the amount of AR terms and $q$ the amount of MA terms on the right hand side. Finding the most optimal ARIMA order $(p, d, q)$ is not trivial (Zhang, 2003;
Aladag et al., 2009). General methods include the Akaike's information criterion (Akaike, 1974) or minimum description length (Rissanen, 1978). However, these methods are often not satisfactory and additional methods have been proposed to determine the order (Al-Smadi and Al-Zaben, 2005). In this article we present results obtained with orders $p = 12$, $d = 1$ and $q = 0$ or $q = 1$, which resulted in good prediction results. Besides, this order avoids including information of past El Niño and La Niña events, which could possibly reduce the prediction skill.
The eventual ARIMA prediction $\hat{Y}_t$ of $\tau = 1$ months ahead is

$$\hat{Y}_t = \sum_{i=1}^{d} Z_{t-i} + \sum_{j=1}^{p} +\alpha_j Z_{t-j} + \sum_{k=1}^{q} \beta_k \varepsilon_{t-k}. \tag{14}$$

Here $\varepsilon_{t-1} = Z_{t-1} - \hat{Y}_{t-1}$. Let $\tilde{Y}_t$ be the ARIMA prediction of $\tau > 0$ months ahead, by applying Eq. 14 for $\tau$ times and replacing any observation $Z_t$ with the consecutive calculated $\hat{Y}_t$, if $t$ is in the future and $Z_t$ therefore unknown. Similarly, if $q = 1$ and $\tau > 1$ months, the residual is calculated by $\epsilon_{t-1} = \tilde{Y}_{t-1} - \hat{Y}_{t-1}$, since the observed value $Z_{t-1}$ is in the future.
After $\tilde{Y}_t$ is predicted by the ARIMA model, the ANN will be used for the prediction $\tilde{N}_t$, making use of more variables than the NINO3.4 index alone. Deciding which of the variables to use is not a straightforward problem, yet crucial for the eventual prediction. Sometimes a pair of two variables can be compatible in the prediction, but perform poor when applied alone. Other pairs can be redundant and cover important information when used alone, but solely noise is included when used together (Guyon and Elisseeff, 2003). Adding a variable to the attribute set and see if it improves prediction, can only conclude whether
it improves prediction with respect to the old attribute set, not whether the variable is predictive in itself. To determine the attribute set, we consider which variables represent a certain physical mechanism that is important for the ENSO prediction. Besides, it is tested whether the prediction skill is reduced if a variable is dropped out of the attribute set.





Moreover, the attributes should be selected at optimal lead times. Apart from considering the physical mechanisms the variables represent, two methods will help to decide which variables can improve the prediction. First, correlation between the predictor and predictant is a commonly used measure for attribute selection (Hall, 1999). Therefore the Pearson cross-correlation is calculated for the attributes at lag $\tau$ to show the predictability of a time series:

$$R_\tau(p,q) = \max_\tau \left( \frac{\sum_{k=1}^n p(t_k)q(t_k - \tau)}{\sqrt{\left(\sum_{k=1}^n p^2(t_k)\right)\left(\sum_{k=1}^n q^2(t_k - \tau)\right)}} \right). \tag{15}$$

Here $p$ is the predictor, $q$ is the predictant and lag $\tau \leq 64$ weeks such that no information too far in the past is considered.

However, the effect of a variable on ENSO at a short lead time increases the cross-correlation at a longer lead time, due to the effect of autocorrelation (Runge, 2014). To solve this autocorrelation problem, a Wiener-Granger causality F-test (Sun et al., 2014) is performed between all predictors $x_1, \cdots, x_N$ and the predictant at lags $\tau$. Note Granger causality is not the same as a 'true' causality. If the test results in a low p-value, the null hypothesis that $x_i$ does not Granger cause the predictant is rejected at a low significance level (i.e. $x_i$ is more likely to Granger cause the predictant). Notice both the cross-correlation and Wiener-Granger method give us merely an idea of which variables can be used for the prediction at different lead times. Both methods are linear, while the attributes will be used in a nonlinear method.

Finally, the $T \times N$ dataset with selected attributes is used to predict the residual between the ARIMA forecast and the observations in an ANN. Besides using the NINO3.4 sequence itself, the additional attributes can be applied to add important information and improve the prediction.

Generally, only a feed-forward ANN is applied, having a structure without loops. The input variables are linearly combined and projected to the first layer neurons according to (Bishop, 2006):

$$z_j = h \left( \sum_{i=1}^D w_{ji}^{(1)} x_i + w_{j0}^{(1)} \right). \tag{16}$$

Here $z_j$ is the value of the $j$-th neuron of the layer, $w_{ji}^{(1)}$ is the weight between input $x_i$ from neuron $i$ to neuron $j$, where the $(1)$ denotes the first layer. $w_{j0}^{(1)}$ is referred to as the bias. $h$ is the so-called (nonlinear) activation function, essential for incorporating the nonlinearity in the prediction model.

These $z_j$ can again be used as input for a second layer, which can be used for a third layer etc.. Eventually this leads to some output which can be compared with the time series that must be predicted. Using a backward propagating technique, the squared error $\sum_t (y_t - \hat{y}_t)^2$ between the residual we are predicting $y_t$ and the output of the ANN $\hat{y}_t$, will be minimized over the weights for the training set. The optimized function can then be tested on the test set. Initially, some random distribution of weights is used. The ANN part of the prediction will be performed with the toolbox ClimateLearn (Feng et al., 2016).

## 3 Analysis of network properties and selection of ML attributes

In this section, topological properties of CNs are analysed within the ZC model and observations, which leads to specific choices of attributes in the hybrid prediction model.




### 3.1 ZC model results

Weekly spatial-temporal data on a $31 \times 30$ grid in the Pacific region are obtained for forty-five years from the ZC model, to construct the CNs. The first five years are not considered, to discard the effect of the initial conditions. A sliding window approach is used to calculate the network variables. This implies that a different network is calculated at each time, which is

sliding four weeks ahead every time step. For the ZC model, either the thermocline network (from $h$), SST network (from $T$), wind network (from $\tau^x$) or a combination of these are considered for CN construction. Multiple network variables are presented here, containing interesting information about the ZC model, although only one will eventually be used in the observations due to its predictive power.

    Determining how strong noise can excite the ENSO mechanisms in the sub-critical case, or determining whether the feed-

backs sustain an oscillation in the supercritical state, could provide information to increase the prediction skill. Feng (2015) found that the skewness of the degree distribution $S_d$ of the CN reconstructed from SST decreases monotonically with increasing coupling strength $\mu$. Although $S_d$ relates to the climate stability and coupling strength, it does not inform whether the system is in either the supercritical or sub-critical state. Here, we introduce a NetOfNet variable which may represent

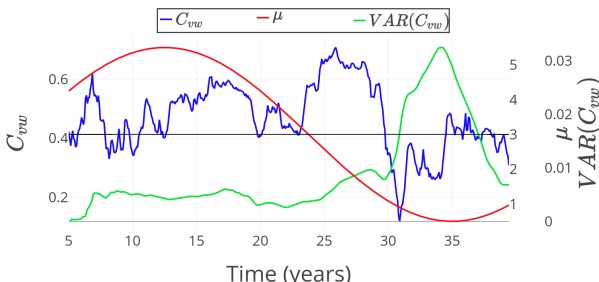

**Figure 2.** Global cross clustering between the SST and wind network in blue and its variance in green in the ZC model. The coupling strength $\mu$ defined as a sinusoid around $\mu_c = 3$ with an amplitude of $0.25$ in red. The sliding window is applied with a window of five years.

properties of the stability of the background state: the global cross clustering ($C_{vw}$) between the SST and wind network. A

sliding window of five years with $\epsilon = 0.6$ was used to compute the networks. In this case, the global cross clustering coefficient is a measure of the amount of triangles in the networks, containing one wind node and two SST nodes. In Fig. 2, this cross clustering is calculated from data from the ZC model, when coupling strength $\mu$ changes periodically in time around the critical value $\mu_c \sim 3.0$. Under sub-critical conditions, the noise has a larger influence on local correlations. This causes triangles to break and the variance of the cross clustering coefficient to increase. The cross clustering $C_{vw}$ is hence a diagnostic network

variable which informs whether the state of the system is in the supercritical or sub-critical regime.

    From the classical view of the oscillatory behavior of ENSO, waves in the thermocline should contain memory of the system, because of their negative delayed feedback. The changing structure of the thermocline network is therefore of interest when predicting ENSO. Calculating this network with threshold $\epsilon = 0.6$ and a sliding window with a length of one year, a zonal pattern in the change of the network close to the equator can be observed during an ENSO cycle. To compare network structures

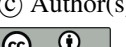



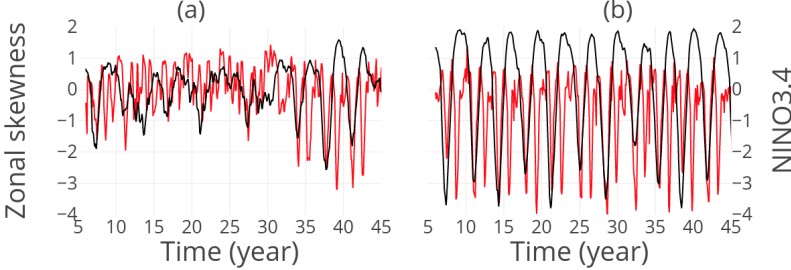

**Figure 3.** Zonal skewness of the degree field of the thermocline network with $\epsilon = 0.6$ and a sliding window of one year in red, NINO3.4 index in black in the ZC model. (a) The sub-critical ($\mu = 2.7$) and (b) the supercritical ($\mu = 3.25$) case.

in the super- and sub-critical state, now constant $\mu = 2.7$ (sub-critical) and $\mu = 3.25$ (supercritical) are taken. Generally, the degree field is quite spatially symmetric, but when ENSO turns either from upward to downward, or from downward to upward, the degree of the nodes in the east decreases. This is at the peak El Niño or La Niña.

To capture this zonal asymmetry around the equator with a variable, the zonal skewness of the degree field will be used
between 7°S to 7°N. The higher the skewness, the more the degree will be located west of the basin. If the skewness is close to zero, the degree is symmetrically distributed over the basin. If it is low, most of the degree is situated in the east. The skewness will show a negative peak when the sign of the first ENSO derivative changes (Fig. 3). In the supercritical case $\mu = 3.25$ this effect is indeed observed. Nevertheless, in the sub-critical case, the pattern is only visible once ENSO shows a clear oscillation (around year 32).

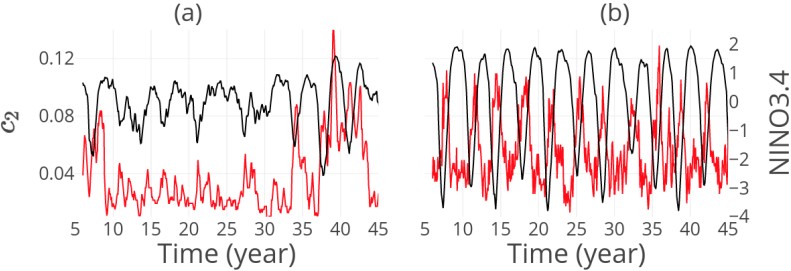

**Figure 4.** The network variable $c_2$ of the thermocline network with a sliding window of one year in red and NINO3.4 in black in the ZC model. (a) The sub-critical ($\mu = 2.7$) case with threshold $\epsilon = 0.99999$ and (b) the supercritical ($\mu = 3.25$) case with $\epsilon = 0.999$.

For the ZC model, $c_2$ (the proportion of nodes belonging to clusters of size two) of the thermocline network is found to indicate the approach to a percolation transition of the network (Fig. 4). Again a window of one year is used. $c_2$ increases approximately one to two years before an El Niño event. This is mainly clear in the supercritical case. In the sub-critical case, a clear warning of an event occurs when the oscillation of ENSO is more clear and the El Niños are stronger. $c_2$ is the only network variable which will not only be applied in the ZC model, but also in the observations later. The quantity $\Delta$ of the same
network behaves similar to $c_2$. Although $\Delta$ does not depend on a chosen threshold like $c_2$, it peaks closer to an El Niño event.





Finally, the algebraic connectivity ($\lambda_2$) can show the spread of information within a network. Specifically, when considering an unweighted NetOfNet from thermocline depth ($h$) and zonal wind ($\tau^x$) with threshold $\epsilon = 0.6$. The spread of information is relatively high before an event, but also after an event, such that $\lambda_2$ peaks both before and after an El Niño event (both for $\mu = 2.7$ and $\mu = 3.25$).

## 3.2 Selecting attributes from observations

The ZC model results have given an indication of the network variables that could be used as attributes in the hybrid model to predict El Niño. Although the network variables show interesting behavior in the ZC model for prediction, this is not always the case in observations. This section describes which variables, including a network variable, are implemented in the hybrid model and the selection of these attributes at different lead times. Notice that only anomalies of the time series in observations are considered.

First, from the recharge/discharge oscillator point of view, the WWV shows great potential for the prediction of ENSO (Bosc and Delcroix, 2008; Bunge and Clarke, 2014). Therefore it is used in the attribute set. The second attribute is a WWV related

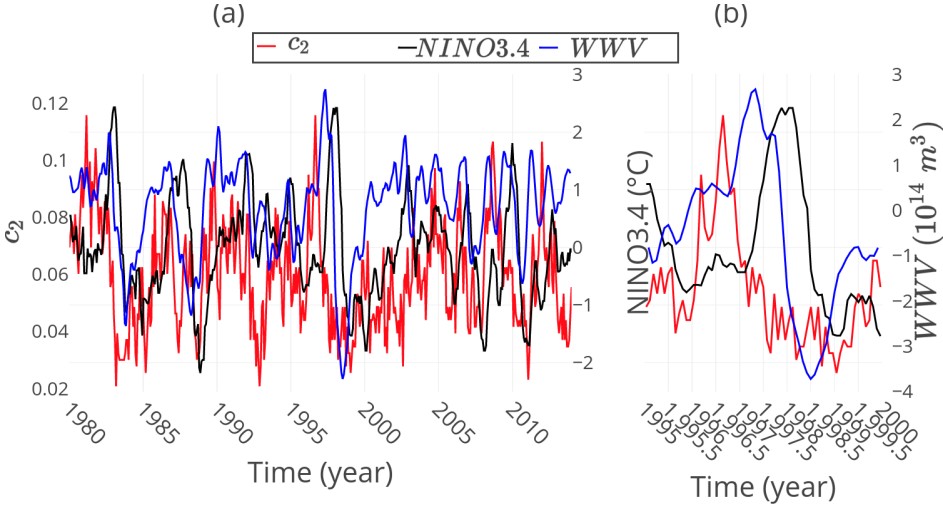

**Figure 5.** The WWV, $c_2$ and the NINO3.4 index from observations for (a) the whole considered time series and (b) only during the 1997 El Niño. A warning of the El Niño event is visible for the WWV and $c_2$. $c_2$ gives a warning almost a year before the 1997 El Niño, while the WWV warns almost seven months ahead.

network variable. The correlations of the SSH time series on a grid of 27 latitude points and 30 longitude points in the Pacific area are used to reconstruct a CN with a threshold $\epsilon = 0.9$ and a sliding window of one year. The sea surface height (SSH) is used instead of thermocline depth, because more data is available and it is by approximation proportional to the thermocline depth (Rebert et al., 1985). During an El Niño event, the link density of this network increases in the warm pool and the cold tongue specifically, causing a percolation-like transition. As discussed in the previous section, an early warning could be obtained with $c_2$. This variable allows us to extend the lead time of the WWV (Fig. 5). Third, atmospheric noise from the





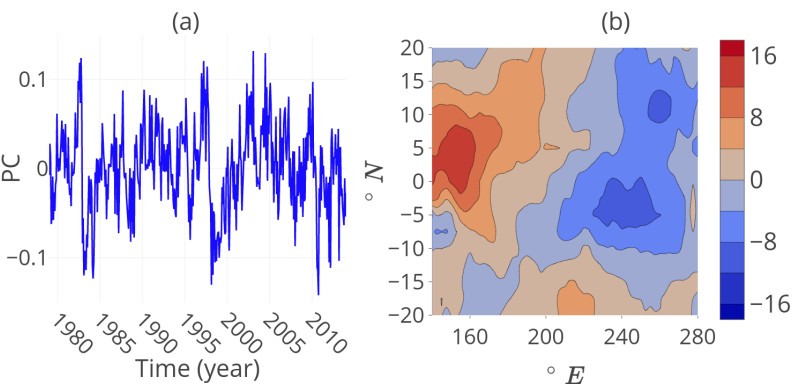

**Figure 6.** (a) The second PC of the residual of the wind stress ($PC_2$) and (b) its EOF, associated with the WWBs.

WWBs are a limitation for the prediction of ENSO (Moore and Kleeman, 1999; Latif et al., 1988). To obtain a variable related to the WWBs, the linear effect of the SST is subtracted from the zonal component of the wind stress. The second principal component ($PC_2$), explaining $8\%$ of the variance, is associated with these WWB's. In Fig. 6, the PC and its EOF are presented. The peaks in the PC are visible before the great El Niño events of 1982 and 1997. Thereby, the EOF has the typical WWB
5    structure, being positive west from the dateline and negative east. Finally, the attribute set does not yet contain any information about the seasonal cycle (SC) yet. The phase locking of an El Niño event to boreal winter is very typical to ENSO. Therefore a sinusoid with the period of a year is used as attribute, to see if it can improve the prediction skill.

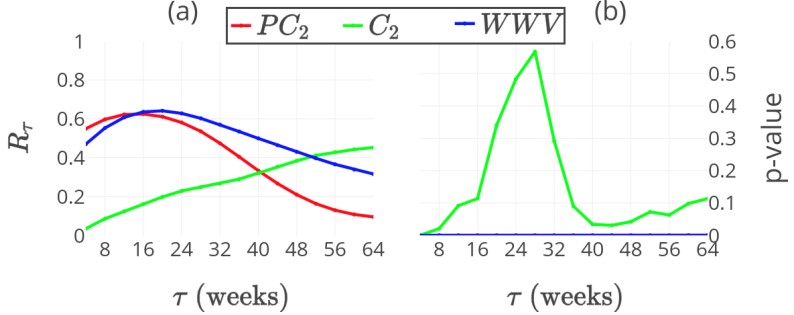

**Figure 7.** (a) The cross correlation of the $PC_2$, WWV and $c_2$ with respect to NINO3.4 for different lags $\tau$. (b) The p-value of the Wiener-Granger hypothesis test for the same lags. A low p-value implies the variable is likely to Granger cause the NINO3.4 index at the specific lag. The p-values of the $PC_2$ and WWV are almost zero for all lags.

     To determine at which lead time the different attributes should be applied, the cross-correlation and the p-value of the Granger test between the attributes and NINO3.4 are considered (Fig. 7). The cross-correlations of $PC_2$ and the WWV show
10    peaks at respectively 12 and 20 weeks, indicating their optimal lead times, since the p-values of the Granger tests are low at every lag and autocorrelation does not play an important role. For $c_2$ however, the cross-correlation increases up to the





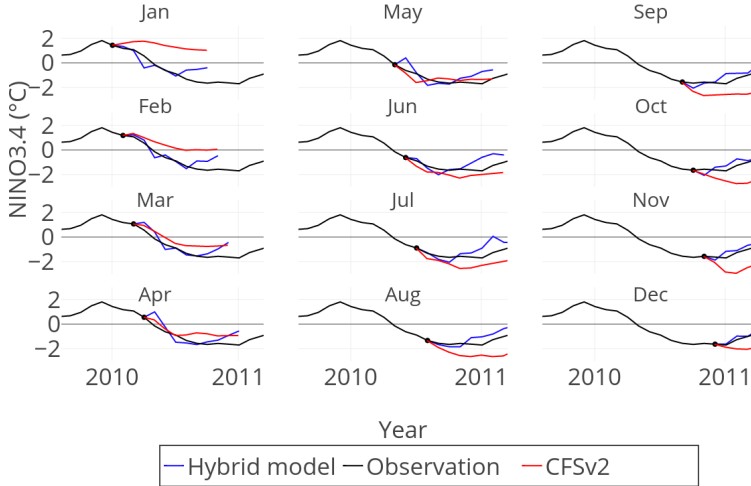

**Figure 8.** Nine-month ahead prediction starting from every month in the year 2010. Blue is the hybrid model prediction with ARIMA(12,1,1), $2 \times 1 \times 1$ ANN structure and attributes are the three-month running mean of WWV, $PC_2$ and SC. The black line is the observed index. Red is the mean of the CFSv2 ensemble prediction.

maximum considered lag, but the p-value of the Granger test has a local minimum close to a lag of 44 weeks. According to these methods, $c_2$ is especially predictive at the longer lead times close to 44 weeks.

To summarize, we are interested in the variables that represent specific physical characteristics related to the prediction of ENSO, to select the attributes. Both $c_2$ and the WWV are related to the recharge/discharge mechanism. $PC_2$ is related to the

atmospheric noise from WWBs. The seasonal cycle (SC) is related to the phase locking of El Niño events to boreal winter. The hybrid model allows us to implement different variables in the attribute set at different lead times. Therefore, the cross-correlations and Wiener-Granger causality were used to determine which attribute is more optimal at various lead times. This showed that it is better to use $c_2$ instead of WWV at lead times of more than 40 weeks. The other network variables which were interesting for the ZC model output (as shown in the previous subsection) are performing worse when applied to observations

and hence are not used as attributes in the hybrid model.

## 4   Prediction results

This section presents the predictions of the hybrid model, as compared with observations and with alternative predictions from the CFSv2 model ensemble of NCEP. The skill with ANN structures up to three hidden layers is investigated. First, a comparison between both predictions is made for the year 2010 (Fig. 8). Moreover, several lead time predictions are shown

and compared to the available CFSv2 lead time predictions. Finally, a recent forecast is made and it is shown how the hybrid model predicts the development of ENSO the coming year.





From now on, the Normalized Root Mean Squared Error (NRMSE) is used to indicate the skill of prediction within the test set:

$$NRMSE(y^A, y^B) = \frac{1}{\max(y^A, y^B) - \min(y^A, y^B)} \sqrt{\frac{\sum_{t_1^{test} \leq t_k \leq t_n^{test}} \left(y_k^A - y_k^B\right)^2}{n}}.$$

Here $y_k^A$, $y_k^B$ are respectively the NINO3.4 index and its prediction at time $t_k$ in the test set. $n$ is the number of points in the

5   test set. A low NRMSE indicates the prediction skill is better.

The year 2010 is a recent example of an under-performing CFSv2 ensemble. Especially in January, all members of the ensemble overestimate the NINO3.4 index, resulting in an overestimation of the ensemble mean (see Fig. 8). The hybrid model is used to predict the same period, with ARIMA(12,1,1) and a $2 \times 1 \times 1$ ANN structure with the three-month running mean of the WWV, $PC_2$, the SC and NINO3.4 itself as attributes. In this case the hybrid model performs better than the CFSv2

10   ensemble. A $2 \times 1 \times 1$ structure means a feed-forward structure with three layers of respectively two, one and one neuron. This ANN structure is found to be the best performing structure at a three-month lead time prediction. It will probably not be the most optimal ANN structure at other lead times.

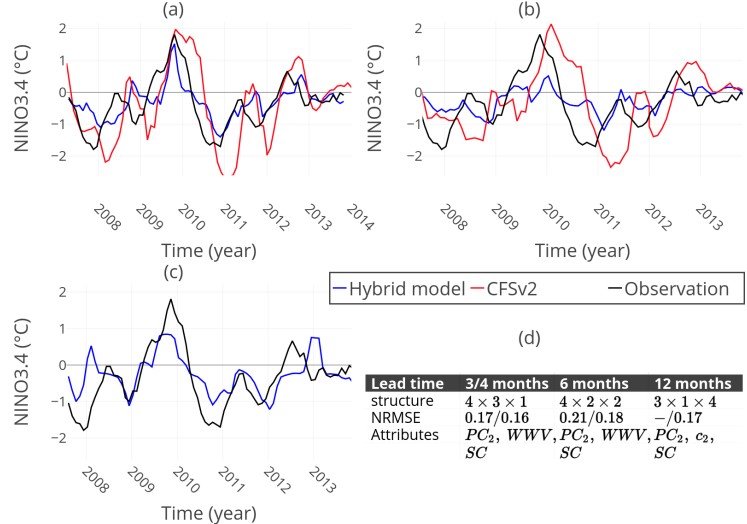

**Figure 9.** NINO3.4 predictions of the CFSv2 ensemble mean (red) and the hybrid model with ARIMA(12,1,0) (blue), compared to the observed index (black). For the hybrid model predictions, from an ensemble of eighty-four different ANN structures, structures resulting in a low NRMSE are presented. (a) The three-month lead time prediction of CFSv2 and four-month lead time prediction of the hybrid model, (b) the six-month lead time predictions and (c) twelve-month lead prediction. The CFSv2 ensemble does not predict twelve months ahead. (d) Table containing information about all predictions: ANN structures of the hybrid model, NRMSEs of the CFSv2 ensemble mean and the hybrid model, and attributes used in the hybrid model predictions.

Considering the three, six and twelve-month lead time predictions, both the three and six-month lead time prediction of the CFSv2 ensemble show some lag and amplification of the real NINO3.4 index (Fig. 9). The hybrid model predictions with





ARIMA(12,1,0) resulting in a low NRMSE and relatively simple ANN structure within an ensemble consisting of eighty-four different ANN structures are also shown in Fig. 9. The eighty-four different structures, are all structures up to three hidden layers with up to four neurons.

Comparing the three-month lead prediction of the CFSv2 ensemble with the four-month lead prediction of the hybrid model,
the lag of the prediction is less and the amplification is not as large in the hybrid model. While the lead time of the hybrid model is one month longer, the prediction skill of the hybrid model is better in terms of NRMSE. The prediction skill of the hybrid model decreases at a six-month lead compared to the four-month lead time prediction. Thereby the lag and amplification of the CFSv2 prediction increase. Although the hybrid model does not suffer as much from the lag, it underestimates the El Niño event of 2010. In terms of NRMSE the hybrid model still obtains a better prediction skill.

To perform a twelve-month lead prediction, the attributes from the shorter lead time predictions are found to be insufficient. However, $c_2$ of the SSH network has shown to be predictive at this lead time, according to its Granger causality and cross-correlation. Therefore the WWV is replaced by $c_2$ for this prediction, which is related to the same physical mechanism. In terms of NRMSE, the twelve-month lead prediction even improves the six-month lead prediction of the hybrid model. On average the prediction does not contain a lag in this period.

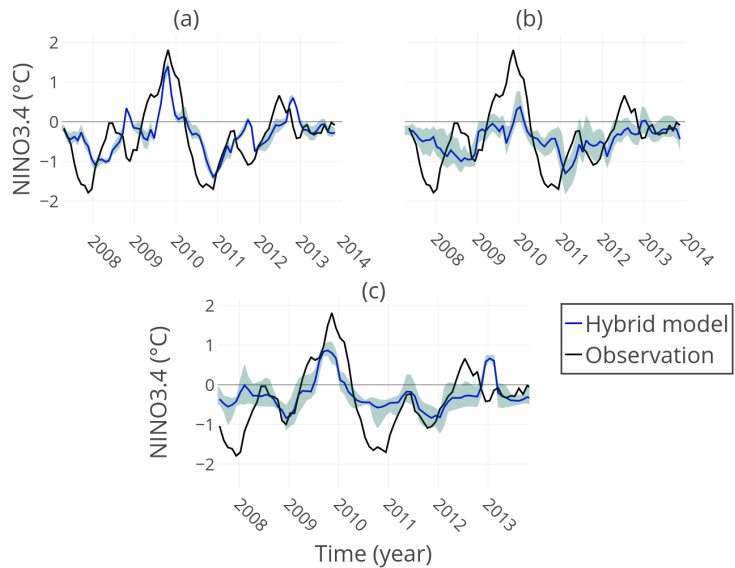

**Figure 10.** Predictions of the NINO3.4 index from an ensemble of hybrid models with different ANN structures and ARIMA(12,1,0) (blue) compared to the observed index (black) with (a) four-month lead time (b) six-month lead time and (c) twelve-month lead time. The ensemble consists of nine predictions with the lowest NRMSE out of eighty-four predictions. The shaded blue area denotes the spread of the nine predictions and the blue line the mean. The NRMSE of the ensemble mean predictions are respectively 0.15, 0.18, 0.17.

To show that the results of Fig. 9 can be generalized, the mean of the predictions with the nine lowest NRMSE of the ensemble with eighty-four different ANN structures is considered (Fig. 10). This ensemble does not differ much from the best





predictions of Fig. 9. The spread of the ensemble remains limited, although it is a bit larger in the six and twelve-month lead prediction compared to the four-month lead prediction.

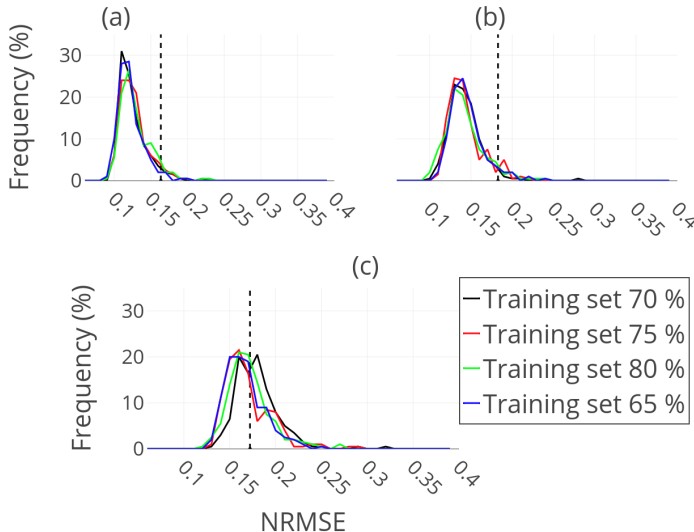

**Figure 11.** Cross validation results of the (a) four, (b) six and (c) twelve-month lead predictions of hybrid models from Fig. 9. Each line presents the frequency every NRMSE is obtained for 200 different initial test sets with a specific training set/test set percentage split . The vertical dashed line denotes the NRMSE of the predictions of Fig. 9.

To test the robustness of these results, a series of cross-validations has been performed. Several percentage splits have been chosen for the training and test set (65-35, 70-30, 75-25 and 80-20), but 200 different initial times of the test set $t_i^{test}$ are randomly chosen between March 1985 and December 2014. This implies that $t_i^{test} > t_f^{train}$ is not necessarily satisfied anymore. If the results for different training and test sets do not deviate much, it is evidence that the model also generalizes to an arbitrary training and test set. The cross validation results of the hybrid models of Fig. 9 are presented in Fig. 11. At all three prediction lead times, the peaks coincide at the same NRMSE for different training-test set ratios. Therefore the different sizes of training and test sets do not seem to influence the result. However, the width of the peaks increases when the prediction lead time increases. This implies the prediction skill becomes more sensitive to the choice of the training and test set in time. Interestingly, at the four and six-month lead time predictions, the average NRMSE is lower than the NRMSE of the prediction of Fig. 9. This implies the predictions with a different training and test set are on average even better than the prediction shown in Fig. 9.

Finally, a prediction is made for the coming year in Fig. 12. Different hybrid models are used at different lead times with ARIMA(12,1,0). ANN structures are chosen that are found to be optimal at the different lead times. For the predictions up to five months, the attributes WWV, $PC_2$ and SC are used from 1980 until present. For the twelve-month lead prediction, the





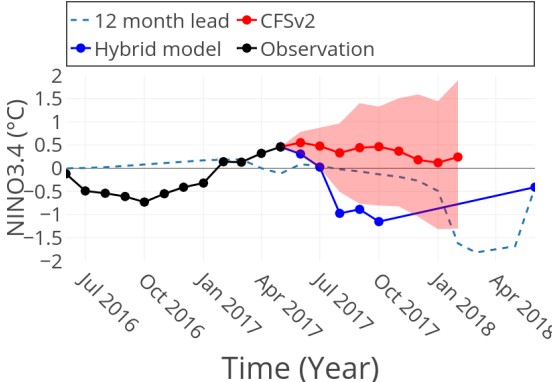

**Figure 12.** NINO3.4 prediction from May 2017. In black the observed index until May 2017. Red is the CFSv2 ensemble prediction mean and the shaded area is the spread of the ensemble. The hybrid model prediction in blue is given by predictions from hybrid models found to be most optimal at the different lead times with ARIMA(12,1,0). The dashed blue line is the running twelve-month lead time prediction.

WWV is replaced by $c_2$ again. This time $c_2$ is computed from the SSALTO/DUACS dataset. Therefore, only a dataset from 1993 until present has been used to train the model and perform the twelve-month lead prediction.

Interestingly, as can be seen in Fig. 12, the hybrid model typically predicts much lower ENSO development than the CFSv2 ensemble. The uncertainty of the CFSv2 ensemble is large, since the spread of predictions is between a strong El

Niño (NINO3.4 index between 1.5 and 2) and a moderate La Niña (NINO3.4 index between -1 and -1.5) for the coming 9 months. The hybrid models predict development to a strong La Niña (NINO3.4 index lower than -1.5) the coming year. From the time of writing, only time will tell which prediction is better. By the time of submission in early March 2018, La Niña conditions are present according to the Climate Prediction Centre of NCEP.

## 5 Summary and Discussion

A successful attempt was made in this paper to use Machine Learning techniques in a hybrid model to improve the skill of El Niño predictions. Crucial for the success of this hybrid model is the choice of the attributes applied to the Artificial Neural Network (ANN). Here, we have explored the use of network variables as additional attributes to several physical ones. Results of the ZC model provided several interesting network variables, such as the cross clustering between wind and SST network, the zonal skewness in the degree field of the thermocline network, and two variables anticipating a percolation-like transition

($\Delta$, $c_2$) and finally the diffusivity of information in the wind and SST network ($\lambda_2$).

Of these network variables, $c_2$ the amount of clusters of size two in a SSH network constructed from observations, is found to provide a warning of a percolation-like transition in the SSH network. This percolation-like transition coincides with an El Niño event. This variable relates to the WWV and hence the recharge/discharge mechanism, but extends the prediction lead time of the WWV when applied in the prediction scheme. Furthermore, apart from both these 'recharge/discharge' related



quantities, the $PC_2$ and SC improve the prediction skill, representing respectively the WWBs and the phase locking of ENSO. The flexibility of implementing different variables at different lead times, allows the hybrid model to improve on the CFSv2 ensemble at short lead times (up to six months). Furthermore, it had a better prediction result than all members of the CFSv2 ensemble in January 2010.

By including the network variable $c_2$, we obtained a twelve-month lead time prediction with comparable skill to the predictions at shorter lead times. This prediction shows a step towards beating the spring predictability barrier. Using ML has the advantage of recognizing the early warning signal of $c_2$ as either a false or true positive. Therefore, it can be a more reliable method then considering a warning when the signal exceeds a certain threshold (Ludescher et al., 2014). Moreover, the early signal from the network variable is not only used to predict an El Niño event, but the development of ENSO, as the hybrid

model provides a regression of the NINO3.4 index. ML serves as a tool which is able to recognize important, but subtle patterns. Something the conventional statistical and dynamical models fail to do in the chaotic system. In the end, the predictions from May 2017 are discussed. By the time of writing, this is the prediction for the coming year. The CFSv2 ensemble mean predicts neutral conditions the coming nine months, with the spread between different members ranging from a strong El Niño to a moderate La Niña. The hybrid model predicts moderate to strong La Niña conditions for the coming year.

Although the results of the methods are promising, some adaptations to the methods attributes still improve predictions. Several network variables resulted in a clear signal in the ZC model, but not necessarily for the observations. Perhaps the cross-correlation and a Granger causality test are not enough to determine the suitability of a variable in the observations. Testing all possible attribute sets in the prediction scheme and comparing results costs time. As a solution, the nonlinear methods 'lagged mutual information' and 'transfer entropy' can be considered techniques to select variables. After all, the

attributes are applied in the nonlinear part of the prediction scheme. Consequently, more variables might be found to increase the prediction skill.

     Even though the currently applied network measures showed interesting properties, different CN construction methods can still be interesting to apply. The Pearson correlation is a simple, effective method to define links between nodes. However, different properties of CNs could be found when using mutual information instead. Moreover, the effect of spatial distance

between nodes can be investigated and corrected for (Berezin et al., 2012). Besides, we have limited ourselves to networks within the Pacific area itself. As ENSO is an important mode in the whole climate system, the area used for CN construction might as well be extended. More specifically, it can be interesting to include the Indian Ocean in the CN construction. Evidence is found that a cold SST in the West of the Indian Ocean is related to a WWB a few months later (Wieners et al., 2016). This could result in a variable related to WWBs, but increasing the lead compared to $PC_2$, which is comparable to $c_2$ increasing the

lead compared of the WWV.

     By applying the ARIMA as simple, yet effective statistical method to apply in the first step of the scheme, the hybrid model shows promising results. However, the exact reason how this model works, remains a topic of investigation. The ARIMA prediction could be related to the linear wave dynamics. It can be interesting to replace the ARIMA part of the scheme by a dynamical model accounting for these linear wave dynamics. For the same reason Vector Autoregression (VAR) can be used





instead of ARIMA. Being a multivariate generalization of an autoregressive model, this can implement the linear effect of other variables on ENSO.

Next to investigation of the exact reason the hybrid model works, some adaptations could still improve the prediction scheme. For example, it is assumed the linear and nonlinear part of the model are additive (see Eq. (10)). This is not necessarily the case

for the real system (Khashei and Bijari, 2011). Besides, the current model does not take into account possible nonlinear effects from the history, since the ANN describes a nonlinear function which does not depend on the history. Moreover, the applied methods searched for a prediction model which is most optimal in terms of least squares minimization. However, it could be interesting to put larger weight at predicting the extreme events in the optimization scheme (as the six-month lead predictions missed the 2010 El Niño event in Fig. 10), or find a function which is more simple (e.g. applying a support vector machine

instead of ANN (Pai and Lin, 2005)).

Although the hybrid model and the attribute selection can clearly be improved, the results here have shown the potential for ML methods, in particular with network attributes, for El Niño prediction. The underlying reason for this success is likely that through the network attributes, more global correlations are taken into account which are needed to be able to overcome the spring predictability barrier.

**Appendix A**

## A1   Pearson correlation

The Pearson correlation between variables $p_i$, $p_j$ associated with two points on a spatial grid $i$, $j \in [0, \cdots, N]$ is defined as:

$$R(i,j) = \frac{\sum_{k=1}^{n} p_i(t_k) p_j(t_k)}{\sqrt{\left(\sum_{k=1}^{n} p_i^2(t_k)\right)\left(\sum_{k=1}^{n} p_j^2(t_k)\right)}}. \qquad (A1)$$

Here $p_i$ is a vector of size $n$ of the time series at time steps $t_k$. The temporal mean is subtracted and data is detrended before

the correlation is calculated.

## A2   Algebraic connectivity

Let $\psi_i$ be the time series at node $i$ in a network. How much $\psi_i$ changes by a hypothetical diffusion process occurring in the network depends on the values of nodes $j$ it is connected to according to the unweighted adjacency matrix $A$ (Newman, 2010):

$$\frac{d\psi_i}{dt} = C \sum_j A_{ij} \left(\psi_j - \psi_i\right). \qquad (A2)$$

Here $C$ is the diffusion constant and $A$ the adjacency matrix. By separating the sum in Eq. (A2), this can be rewritten as:

$$\frac{d\psi_i}{dt} = C \sum_j \left(A_{ij} - \delta_{ij} d_i\right)\psi_j, \qquad (A3)$$



$\delta_{ij}$ is the Kronecker delta function and $d_i$ the degree of node $i$. In matrix notation this reduces to:

$$\frac{d\psi_i}{dt} = C\left(\mathbf{A} - \mathbf{D}\right)\psi. \tag{A4}$$

Here $\mathbf{D}$ with $\mathbf{D_{ij}} = \delta_{ij}\mathbf{d_i}$ is a square matrix that contains the degrees at the diagonal and zero elsewhere. Now we define the graph Laplacian of the network as:

5  $$\mathbf{L} = \mathbf{A} - \mathbf{D}. \tag{A5}$$

Equation A4 reduces to the diffusion equation, but with the graph Laplacian matrix $\mathbf{L}$ instead of $\nabla^2$. By calculating the eigenvalues of the Laplacian matrix $\lambda_1, \cdots, \lambda_n$ with $\lambda_1 \leq \lambda_2 \leq \cdots \leq \lambda_n$, we can determine the diffusion within the network. Since the matrix is symmetric, the eigenvalues are real. Moreover, the smallest eigenvalue $\lambda_1$ is always zero and no eigenvalues are negative. This means $\psi_i$ will decay to a stable solution. The second smallest eigenvalue, called the algebraic connectivity,

10  is of particular interest. In general, $\lambda_2 > 0$ if the network has a single component.

*Acknowledgements.* PN would like to thank the *Instituto de Física Interdisciplinar y Sistemas Complejos* (IFISC), for hosting his stay in Mallorca during part of 2017.

CL and EHG acknowledge support from Ministerio de Economia y Competitividad and Fondo Europeo de Desarrollo Regional through the LAOP project (CTM2015-66407-P,MINECO/FEDER)



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
