# Peer review of "Using Network Theory and Machine Learning to predict El Niño"

_Earth System Dynamics, 2018_

## Referee Comment (RC1) · R. Link (Referee) · 28 Mar 2018

**General comments**

In "Using Network Theory and Machine Learning to predict El Niño" the authors develop a predictive model for the NINO3.4 index of El Niño strength. The model uses network theory to select a set of predictors to use in the regression. The predictions are generated by summing an ARIMA function with the output of a neural network with the predictors as inputs. This design can be thought of as an autoregressive extrapolation of trends in the time series, modified by modified by shocks forecast by the predictors. This model design is an interesting and innovative approach to the problem. However, the paper suffers from several major flaws that call the results into question.

[Figure]

The first is the unusual design of the cross-validation calculation. The initial description on p. 7 of the separation into training and testing sets is standard, and the authors make an important point:

> Note that, since we are predicting time series, for any training set $[t_i^{train}, t_f^{train}]$ and test set $[t_i^{test}, t_f^{test}]$, $t_i^{test} > t_f^{train}$ must hold...

This is entirely correct, but on p. 16 the authors acknowledge that they violate this condition in their cross-validation experiment. Additionally, in that same section they appear to treat cross-validation calculations with different relative sizes of testing to training sets, run on the *same* dataset as independent cross-validation experiments, which they definitely are not. Together, these factors render the entire cross-validation exercise highly questionable, particularly where the results depicted in Figure 11, and any conclusions derived from them, are concerned. In particular, it seems likely that the peaks in Figure 11 are a reflection of the fact that many of the testing sets used in the result overlap with the training set, and not a realistic estimate of the model's likely performance out of sample.

A related problem is the paper's treatment of hyperparameter tuning. The authors do not provide a list of the hyperparameters used in the model, but certainly the $p$, $q$, and $d$ parameters of the ARIMA model qualify, as do the number and sizes of the neural network layers. Possibly the choice of predictors and their lead times are another set of hyperparameters, although possibly not, if they were chosen exclusively based on the Z-C model results. The paper is vague on this point, but several passages, such as this one:

> Deciding which of the variables to use is not a straightforward problem, yet crucial for the eventual prediction. Sometimes a pair of two variables can be compatible in the prediction, but perform poorly when applied alone.....

suggest that the predictor choice was tuned using the data. Indeed, the entire subject of how the hyperparameters were tuned is not discussed at all. This, combined with the problems with the cross-validation, suggests that the tuning of hyperparameters is likely to have caused substantial overfitting in the model.

I also found it rather difficult to understand the intended operation of the model. One might expect that the model is meant to be applied starting at some $t = t_0$ and working forward step by step, presumably with the model fidelity degrading the further the forecast is pushed into the future. However, the paper presents a family of three models tuned for different lead times, each with different model structures, and in one case different predictors. Since each model can make a forecast at any future time by either extending the forecast (for the short lead time models) or by using the intermediate steps from equation (14) (for the long lead time models), it is not clear how these variants on the model are meant to be reconciled. It's possible that they are intended to be averaged or used in some other boosting procedure, but if so, this is not adequately explained.

Finally, the paper's confusing structure makes it very difficult for readers to work out the exact details of the modeling and validation procedures. Much seemingly irrelevant information is included, some important information is left out, and detailed explanations are often deferred until later in the paper, well past when the topics they pertain to are introduced. A major contributor to this confusion is the bottom-up organization of the paper. Calculations are introduced early in the discussion without context (and sometimes, as in §2.3, without even a clear indication of what variables the calculations are being applied to). Later on, these calculations are assembled into a final product, but in the meantime readers are left with little guide as to why the constituent calculations are being done a certain way, which calculations are significant and which are merely asides, how the pieces being described will eventually fit together, and so on. The paper would be a lot clearer if it provided more context early in the discussion, so that readers can more easily understand what role each of these calculations will eventually

play in the final model.

**Specific comments**

At no point are we ever told what activation function was used for the neural networks.

In Figure 9 on p. 14 the NRMSE loss function for the three variants of the model compared to the corresponding figures for the CFSv2 ensemble mean. The loss values quoted in the figure are:

| Lead time | CFSv2 loss | Hybrid model loss |
|---|---|---|
| 3–4 mo. | 0.17 | 0.16 |
| 6 mo. | 0.21 | 0.18 |
| 12 mo. | N/A | 0.17 |

Is the reported difference between the Hybrid model and the CFSv2 a substantial improvement? The performance of the 3–4 month models looks nearly equivalent, and even the 0.03 difference in NRMSE for the 6 month model looks likely to be within the range of variation in the models' performance over different datasets. What argument can the authors make to support the idea that this model will produce materially better ENSO predictions than existing models?

Section 2.2, covering the Zebiak-Cane model goes into a lot of detail that doesn't seem strictly germane to how the Z-C model will be used in the construction of the predictive model. On the other hand, the single most important detail, namely, the outputs of the Z-C model that will be used in the construction of the predictive model, is omitted. This section also gives a lengthy discussion of a procedure for adding noise to the Z-C results, but the purpose of adding this noise is not explained.

In the introduction there is a reference to the Alpha Go project. This isn't really relevant to the topic of this paper. First of all, the neural networks used in Alpha Go are much

more complex than the ones used here. Second, the tasks they are being asked to perform are quite different from the task described here. Therefore, the success of the neural networks in that project doesn't tell us much about what kind of success we might expect in this application.

Appendix A seems a little extraneous. A.1 is a restatement of the equation for the Pearson correlation coefficient. This statistic is well-known, and its definition need not be repeated here. The statistic in A.2, on the other hand, does merit description, but it is not clear what it is actually used for in the analysis. It seems to be mentioned at the end of section 2.3 and then not used again.

---

## Referee Comment (RC2) · Anonymous Referee #2 · 13 Apr 2018

Overall I think this is valuable and important work, but I think there could be more clarity in the writing. It tends to read as a long sequence of sentences rather than a narrative that walks the reader through the steps of the analysis. At the end, I'm left slightly confused as to (i) how did you use the CZ model; did you actually learn something from that that helped analyze the real world, (ii) how you decided on the specific input variables (rather than what sounds like a jumbled mess of exploring a wide variety of different concepts that might have some relevance), (iii) to what extent your improvement in prediction is actually related to ML/ANN versus having identified good predictive variables (e.g., could you have identified a linear model that used those variables and obtained a good prediction? Were the ultimate relationships "learned" by the ANN between inputs and output actually notably nonlinear?), and (iv) it would help

to have a single final plot showing rms error vs prediction horizon as compared with the current methods.

1. P2, 1st line, not quite sure how to define "intuition and creative thinking", nor (more importantly) why this is relevant here.

2. P2, par lines 3-11, this seems a bit awkwardly worded. It isn't a binary choice between many layers and inputs and "simpler", but rather a continuum of choices with an inherent trade-off. Using more layers and input variables means you can rely more on the algorithm to figure out what matters at the expense of needing to train it on more training data, and the fewer variables/layers one uses the less training data might be required but the more that forces the user to make intelligent choices for input variables rather than relying on the algorithm to do so.

3. The choices in Section 2.3 are not well motivated (that is, why are these the relevant choices to feed into the ANN, and what else did you try?) This section could benefit from a couple of introductory sentences that describe the goal of the section, and the broad overview of the ideas of the section.

4. Why is it adequate to have all of the memory embedded in the linear part of the model?

5. For that matter, not entirely obvious to me, since you are using ML to predict the nonlinear terms anyway, whether the ML can also predict the linear (but dynamic) part without any extra effort, or for that matter the nonlinear and dynamic part. Did you try different things and conclude you didn't have enough training data to converge, and kept simplifying, or did you just guess what might work and then it did? I didn't go back and read Hibon and Evgeniou, but it would seem like the question of how to simplify what the ML is actually learning is case dependent rather than absolute. Some more motivation here is required (and at a minimum you should clarify what is meant by "more stable" and provide a few more words of intuition as to why this reduces the risk of a bad prediction.)

6. Extra plus sign in eqn 13 and 14. Also, shouldn't the summation on the second term start at d+1 (otherwise, the j=1 in the second term and the i=1 in the first term are identical, and you have a standard ARMA model rather than an ARIMA model). (Also, don't recall if you said why you were using ARIMA rather than ARMA?)

7. P7, L19-20, why would including past El Nino and La Nina information reduce prediction skill?

8. P8, L1, I'd have just thought the choice of lead time is like a choice of different variables, that there's nothing wrong with including the same variable at different times as part of the input.

9. P8, L17, "generally" as in, "in this paper", or "generally" as in "in most research"?

10. Section 3.1, any reason why you only used 45 years of ZC output? Why not use a few thousand years of output? (I ran it for that long quite a long time ago, so I know it isn't a computational challenge to do.)

11. Also, section 3.1, you might want to say up front a bit more about motivation – are you trying to learn from ZC which variables are best to use, or ultimately comparing predictive capability on ZC vs the real world, or get a good initial estimate of ANN weights from ZC so that you don't have to converge as much when you apply to the real world? These are all possible goals, but other than the second one, may be problematic if the physics in ZC doesn't match the real world physics (and while with their original parameter choices the equilibrium point in ZC is unstable with a chaotic self-sustained response, I think the general consensus now is that the real world isn't exhibiting chaos but rather stochastically forced response of a damped stable system). This is similar to the comment on Section 2.3; it would be helpful to have a few additional sentences that talk about where you're going with a section, why is it here, what are you hoping to learn, and what the structure of the section is. (I note subsequently that you never actually look at the predictability of CZ model, improvement thereof with ANN, and you also don't use the same variables in the real world analysis… can you be clear as to

why this section is here and what you learned? Is it here just because you spent a lot of time on it and figure that should be documented somewhere, or is it essential to motivate the analysis of the real world?)

12. P10, L2, I think what you mean here is something like "when the ENSO index changes from increasing to decreasing (peak El Nino) or from decreasing to increasing (peak La Nina)"? (The wording is a bit unclear to me.) Similarly line 7, refer to the derivative of the ENSO index, rather than the derivative of ENSO… (to me, "ENSO" refers to the overall dynamic phenomenon, which isn't a thing that has a sign or a derivative).

13. Section 4, rather than just focusing on a few things like 2010 (which is cherry-picked as a year where the default scheme does badly), and a few prediction horizons, one thing that would help evaluate this method would be a single plot of rms prediction error versus time for the two methods (that is, for any month once you have sufficient past data, do the N-month prediction for every N up to a year or more using both methods, and then over this big set of month N predictions, what's the rms error?) This would also be a great way to compare your ARIMA alone with ARIMA + ANN.

14. P14, L11, what do you mean by "best-performing"? What metric? Does that mean that adding more neurons made it worse? Or do you just mean that adding more neurons didn't make it better?

15. P15, L4, why compare the two methods at different lags instead of the same lag?

16. P15, L7, doesn't this contradict the abstract?

17. P15, L14, I'm confused by this sentence – you do a better job at predicting things 1 year in advance than 6 months?

18. Also, I must have missed something; I thought you'd already picked the set of input variables, and now it sounds like you are only using a subset, and a different subset for each prediction horizon. Overall, this sounds incredibly fragile. You do a lot of work to

pick a few really good input variables, and any time you change the time horizon you might need to change those, and change the number of neurons... I thought the whole point of ANN was the ability to be lazy and let the algorithm do all the work for you!

19. P15, L15-16, again, I'm a bit confused... why do we need to maintain a whole ensemble of different ANN structures? This doesn't converge to something with enough neurons? Also, Figure 11, am I interpreting this right that you found a bunch of possible ANN structures that outperform the ones in Figure 9? (Sorry, I'm totally lost at this point so this might be off-base and simply imply some insufficient description.) Why not go back and redo Fig 9 with the better ANN structure? This entire section reads a bit as a collection of odds and ends of results rather than as a post-facto summary.

---

## Author Comment (AC1) · 6 Jun 2018

**General Comments**

*In 'Using Network Theory and Machine Learning to predict El Niño' the authors develop a predictive model for the NINO3.4 index of El Niño strength. The model uses network theory to select a set of predictors to use in the regression. The predictions are generated by summing an ARIMA function with the output of a neural network with the predictors as inputs. This design can be thought of as an autoregressive extrapolation of trends in the time series, modified by modified by shocks forecast by the predictors. This model design is an interesting and innovative approach to the problem. However, the paper suffers from several major flaws that call the results into question.*

We would like to thank Robert Link for his careful reading and his constructive comments.

Please find our replies and the points that will be changed in the revised manuscript below.

On behalf of all the authors,

Peter Nooteboom

**1   Major Comments**

*1. The first is the unusual design of the cross-validation calculation. The initial description on p. 7 of the separation into training and testing sets is standard, and the authors make an important point:*

> *Note that, since we are predicting time series, for any training set $[t_i^{train}, \ t_f^{rain}]$ and test set $[t_i^{test}, \ t_f^{test}]$, $t_i^{test} > t_f^{train}$ must hold...*

*This is entirely correct, but on p. 16 the authors acknowledge that they violate this condition in their cross-validation experiment.*

**Author's response**

Most of the results in the manuscript do satisfy the constraint $t_i^{test} > t_f^{train}$ above (see figures 8, 9 , 10, 12 of the old manuscript). To satisfy the constraint is convienient in these results, from the intuitive idea that the model is first trained on all data in the past to make a real prediction in the future, as is done in Fig. 12 (which is not a hindcast). It would be more clear if we state here that this condition 'is convenient' in stead of 'must hold.'

However, for the cross-validation method in Fig. 11 (enumeration in the previously submitted version), it is difficult to meet this condition, since the observational time series are too short. As stressed in [1], a cross-validation which only considers a last block such as in figures 9 and 10 (enumeration in the previously submitted version), does not make full use of the data. For the validation method of Fig. 11 we follow Ref. [1] in which it is empirically shown, and justified, that violating the constraint $t_i^{test} > t_f^{train}$ could be acceptable in some cases and lead to an improved performance. Another motivation for this cross-validation method is that asymptotic behavior from theory might behave differently on small test sets. Nevertheless in the rest of our calculations we respect $t_i^{test} > t_f^{train}$.

**Changes in manuscript**

We will change 'must hold' at page 7, line 6 into 'is convenient.'
We will include reference [1], and we will explain why we chose this type of cross-validation in one of the calculations in the revised manuscript.

*2. Additionally, in that same section they appear to treat cross-validation calculations with different relative sizes of testing to training sets, run on the same dataset as independent cross-validation experiments, which they definitely are not.*

**Author's response**

Thank you for mentioning this point. The cross-validation experiments with different relative sizes are presented to check if the size of the training and test set matters. One might expect that a shorter training set could decrease the prediction skill, simply because there is less data for the model to train. This means that different percentage splits could overlap in time. However, it is true that the manuscript should contain an explanation on why the different relative sizes of training and test sets are considered.

**Changes in manuscript**

In the revised manuscript it will be explained why the different relative sizes of training and test sets are considered in the cross-validation.

*3. Together, these factors render the entire cross-validation exercise highly questionable, particularly where the results depicted in Figure 11, and any conclusions derived from them, are concerned. In particular, it seems likely that the peaks in Figure 11 are a reflection of the fact that many of the testing sets used in the result overlap with the training set, and not a realistic estimate of the model's likely performance out of sample.*

**Author's response**

From the previous two comments it is clear that we use this type of cross-validation in this particular figure to make full use of the available data, as explained in Ref. [1]. Also, the objective of this figure is to show the stability of the method with different sizes of the training and testing sets.

**Changes in manuscript**

In the revised manuscript it will be explained why this type of cross-validation method is chosen.

*4. A related problem is the paper's treatment of hyperparameter tuning. The authors do not provide a list of the hyperparameters used in the model, but certainly the p, q, and d parameters of the ARIMA model qualify, as do the number and sizes of the neural network layers. Possibly the choice of predictors and their lead times are another set of hyperparameters, although possibly not, if they were chosen exclusively based on the Z-C model results. The paper is vague on this point, but several passages, such as this one:*

> *Deciding which of the variables to use is not a straightforward problem, yet crucial for the eventual prediction. Sometimes a pair of two variables can be compatible in the prediction, but perform poorly when applied alone.. . .*

*suggest that the predictor choice was tuned using the data. Indeed, the entire subject of how the hyperparameters were tuned is not discussed at all.*

**Author's response**

The ANN structure is indeed tuned on the data. Therefore, besides the cross validation, Fig. 10 is included to show that this structure can be generalized and more structures lead to a similar result, which is evidence that they converge to a similar function from predictor to predictant.

The order of the ARIMA(p,d,q) model is not tuned. We just present the results where $p = 12$ to consider information up to a year ahead, with which we already obtain good results.

The choice of the predictors was mainly based on the ZC-model results which identify the physical reasons that would lead to a good prediction. This improved the search for attributes which would contain important information for prediction, but remain relatively independent. By choosing them at a specific lag, also their performance, cross-correlation and Wiener-Granger causality with the NINO3.4 index is considered, which could lead to the replacement of physically related variables.

**Changes in manuscript**

We will follow the suggestion to explicitly name the hyperparameters which have to be tuned for the model in the revised manuscript, and explain how these are tuned. This will done at the end of section 2.4. The hyperparameters which are named are correct and we will give an explanation of the tuning for these different hyperparameters in the revised manuscript.

In the revised manuscript, we will add the spread of hybrid models with different $p$ of the ARIMA order, to show that the predictions do not vary much in this range of ARIMA orders.

*5. This, combined with the problems with the cross-validation, suggests that the tuning of hyperparameters is likely to have caused substantial overfitting in the model.*

**Author's response**

We show that the prediction is not very sensitive to the hyperparameters which are tuned on the data (the ANN structure and the ARIMA order). The test sets of Fig. 9 and 10 already provide some evidence that the model is not overfitting and the applied cross-validation method shows that the prediction model does not depend on different training and test sets. Nevertheless, we still cannot completely rule out overfitting outside the available data we have. Even if there is a chance the model is overfitting outside the available data we have, we think the proposed approach is still interesting for prediction of ENSO. Note that more studies about El Niño prediction have troubles with the shortness of the available time series [2] and overfitting will always be a possibility.

**Changes in manuscript**

We will include reference [2] in the discussion of the revised manuscript and explain it is difficult to rule out that the model is overfitting because of the short time series.

*6. I also found it rather difficult to understand the intended operation of the model. One might expect that the model is meant to be applied starting at some $t = t_0$ and working forward step by step, presumably with the model fidelity degrading the further the forecast is pushed into the future. However, the paper presents a family of three models tuned for different lead times, each with different model structures, and in one case different predictors. Since each model can make a forecast at any future time by either extending the forecast (for the short lead time models) or by using the intermediate steps from equation (14) (for the long lead time models), it is not clear how these variants on the model are meant to be reconciled. It is possible that they are intended to be averaged or used in some other boosting procedure, but if so, this is not adequately explained.*

**Author's response**

The hybrid models at the different lead times are independent of each other. Part of the approach is that we tuned the model at specific lead times, to find which configuration is better for the memory contained in the attributes. That is also why we have different attributes at different lead times. This also means that, if we find more attributes via network analyses in future research which contain different length of memory, these attributes can be applied at the different lead times. This allows us to tune the hybrid model at different lead times.

**Changes in manuscript**

In the revised manuscript, we make clear that these hybrid models are tuned independently from each other and do not 'start at some $t = t_0$ and work forward step by step' (Sect. 2.4).

*7. Finally, the paper's confusing structure makes it very difficult for readers to work out the exact details of the modeling and validation procedures. Much seemingly irrelevant information is included, some important information is left out, and detailed explanations are often deferred until later in the paper, well past when the topics they pertain to are introduced. A major contributor to this confusion is the bottom-up organization of the paper. Calculations are introduced early in the discussion without context (and sometimes, as in §2.3, without even a clear indication of what variables the calculations are being applied to). Later on, these calculations are assembled into a final product, but in the meantime readers are left with little guide as to why the constituent calculations*

*are being done a certain way, which calculations are significant and which are merely asides, how the pieces being described will eventually fit together, and so on. The paper would be a lot clearer if it provided more context early in the discussion, so that readers can more easily understand what role each of these calculations will eventually play in the final model.*

**Author's response**

The reason for the current structure of the paper is that it includes part of the process of how we got to the attributes applied in the hybrid model. We tried to find a physical reason for the variables to be included in the attribute set of the hybrid model, such that it increases the probability of a good prediction. To do this we looked at the dynamics of the ZC model and applied a network analyses to this model. We found some interesting attributes from this network analysis, but most of them were eventually not applied in the prediction model, because they did not behave similar when using observations. We understand this can be of confusion for the reader.

**Changes in manuscript**

As a solution, the results which are not used in the hybrid model (that is everything in section 2.3 and 3.1 which is not related to the attribute $c_2$ which is applied in the hybrid model) will be put in an appendix. Hopefully, this will establish a better connection between the results from the ZC model and the part about the hybrid model.

**2    Specific comments**

*1. At no point are we ever told what activation function was used for the neural networks.*

**Author's response**

The activation function used is the Sigmoid function.

**Changes in manuscript**

We will add this information in the revised manuscript.

*2. In Figure 9 on p. 14 the NRMSE loss function for the three variants of the model compared to the corresponding figures for the CFSv2 ensemble mean. The loss values quoted in the figure are:*

| Lead time | CFSv2 loss | Hybrid model loss |
|-----------|-----------|-------------------|
| 3-4 mo. | 0.17 | 0.16 |
| 6 mo. | 0.21 | 0.18 |
| 12 mo. | N/A | 0.17 |

*Is the reported difference between the Hybrid model and the CFSv2 a substantial improvement? The performance of the 3-4 month models looks nearly equivalent, and even the 0.03 difference in NRMSE for the 6 month model looks likely to be within the range of variation in the models' performance over different datasets. What argument can the authors make to support the idea that this model will produce materially better ENSO predictions than existing models?*

**Author's response**

It is true that the hybrid model performs better than the CFSv2 ensemble mean at the shorter lead times, but we do not consider this to be the important result in the the table displayed in the figure. Up to six months ahead, the predictions are known to be quite good nowadays [3]. The most important result we find is that the twelve month lead prediction performs similar or even better than the shorter lead time predictions because of the attributes we chose and hence it is breaking the spring predictability barrier.

**Changes in manuscript**

In the revised manuscript we will put more emphasis on the important result that the twelve month lead prediction performs similar or even better than the shorter lead time predictions.

*3. Section 2.2, covering the Zebiak-Cane model goes into a lot of detail that doesn't seem strictly*

*germane to how the Z-C model will be used in the construction of the predictive model. On the other hand, the single most important detail, namely, the outputs of the Z-C model that will be used in the construction of the predictive model, is omitted. This section also gives a lengthy discussion of a procedure for adding noise to the Z-C results, but the purpose of adding this noise is not explained.*

**Author's response**

An important purpose of the ZC-model is to explain the main dynamics which is associated with ENSO. This is used to find good attributes for the hybrid model. That is why the network analyses is first applied to the ZC model, resulting in a network variable $c_2$, which is eventually used in the hybrid model.

Noise is introduce as a way to model high-frequency atmospheric variability. The effect of adding the noise is explained on p. 4 of the manuscript:

> The effect of the noise on the model behavior depends on whether the model is in the super- or subcritical regime (i.e whether $\mu$ above or below $\mu_c$). If $\mu < \mu_c$, the noise excites the ENSO mode, causing irregular oscillations. In the supercritical regime, a cycle of approximately four years is present, and noise causes a larger amplitude of ENSO variability.

Hence the noise can excite the ENSO variability and can be an important factor for the prediction of ENSO. This leads to the reason for including the second principal component of the residual of the wind stress (PC2) in the attribute set (see p. 12).

**Changes in manuscript**

We make the purpose of the ZC model more clear in the revised manuscript.

*4. In the introduction there is a reference to the Alpha Go project. This isn't really relevant to the topic of this paper. First of all, the neural networks used in Alpha Go are much more complex than the ones used here. Second, the tasks they are being asked to perform are quite different from the task described here. Therefore, the success of the neural networks in that project doesn't tell us much about what kind of success we might expect in this application.*

**Author's response**

The Alpha Go project indeed made use of different type of machine learning.

**Changes in manuscript**

We will delete this citation and do not mention the project anymore in the revised manuscript.

*5. Appendix A seems a little extraneous. A.1 is a restatement of the equation for the Pearson correlation coefficient. This statistic is well-known, and its definition need not be repeated here. The statistic in A.2, on the other hand, does merit description, but it is not clear what it is actually used for in the analysis. It seems to be mentioned at the end of section 2.3 and then not used again.*

**Author's response**

The statistic $\lambda_2$ in Appendix A2 is computed from the ZC model in section 2.3.

**Changes in manuscript**

We will remove Appendix A1 from the old manuscript as suggested.

Part of section 2.3 and 3.1 of the old manuscript is not used in the hybrid model. We will move these parts to the appendix. This means that the part of section 2.3 that will be moved to the appendix will become appendix A1, and the part of section 3.1 that wiil be moved to the appendix becomes appendix A2. As a consequence, the statistic $\lambda_2$ is explained in the same section as other Climate Network properties which are applied to the ZC model (but not used in the hybrid model). This makes the purpose of $\lambda_2$ more clear. We will make the explanation of how the variable $\lambda_2$ is calculated shorter, since it can also be found in [4].

**References**

[1] Christoph Bergmeir and José M. Benítez. On the use of cross-validation for time series predictor evaluation. *Inf. Sci. (Ny).*, 191:192–213, 2012.

[2] Wasyl Drosdowsky. Statistical prediction of ENSO (Nino 3) using sub-surface temperature data. *Geophys. Res. Lett.*, 33(3):10–13, 2006.

[3] L. Goddard, S. J. Mason, S. E. Zebiak, C. F. Ropelewski, R. Basher, and M. A. Cane. Current approaches to seasonal-to-interannual climate predictions. *Int. J. Climatol.*, 21(9):1111–1152, 2001.

[4] M.E.J. Newman. *Networks: An introduction*, volume 6. Oxford university press, Oxford, 2010.

---

## Author Comment (AC2) · 6 Jun 2018

We would like to thank the referee for his careful reading and his/her constructive comments.

Please find our replies and the points that will be changed in the revised manuscript below.

On behalf of all the authors,

Peter Nooteboom

*Overall I think this is valuable and important work, but I think there could be more clarity in the writing. It tends to read as a long sequence of sentences rather than a narrative that walks the reader through the steps of the analysis.*

**Reply**

Thank you for pointing this out. We think that the reason that the current structure could be somewhat confusing, is that a lot of network variables are explained in the beginning, and are not used anymore later in the hybrid model (except for one). We think the results from the network analyses of the ZC model are interesting.

**Changes in manuscript**

We will move the network variables which are eventually not used in the hybrid model to the appendix. As a result the paper will read more as 'a narrative that walks the reader through the analyses.'

*At the end, I'm left slightly confused as to (i) How did you use the CZ model; did you actually learn something from that that helped analyze the real world,*

**Reply**

The attributes which are applied in the hybrid model all represent a physical process. The first reason the ZC model is presented, is that it represents these physical processes which are important for prediction (e.g. the atmospheric noise that excites the ENSO mode is a reason to add PC2 in the attribute set). Second, we applied an analysis on the ZC model using Network Theory. This leads to multiple variables, of which one also showed interesting properties in observations and is applied in the attribute set.

**Changes in manuscript**

In the revised manuscript we will add an additional motivation to use the ZC model in section 2.2.

*(ii) How you decided on the specific input variables (rather than what sounds like a jumbled mess of exploring a wide variety of different concepts that might have some relevance)*

**Reply**

First the ZC model is used to investigate which variables could be interesting to apply from a physical point of view and the network analyses is applied on the model to find variables which contain useful information for prediction. Then the cross-correlation and Wiener-Granger causality are calculated at the different lags to see which of the variables we could apply at the different lead times. Finally, those variables are used in the hybrid model to see how good the prediction performed and it is tested if they are also robust at different training and test sets.

**Changes in manuscript**

In order to avoid the 'jumbled mess,' we will move everything related to ZC model results which are not directly used as input in the ANN to the appendix of the revised manusscript.

*(iii) To what extent your improvement in prediction is actually related to ML/ANN versus having identified good predictive variables (e.g., could you have identified a linear model that used those variables and obtained a good prediction? Were the ultimate relationships "learned" by the ANN between inputs and output actually notably nonlinear?)*

**Reply**

ANN is known to be a good tool for prediction in nonlinear systems, such as the ENSO system. The ANN can recognise patterns which are important for prediction, which could be missed by the conventional statistical models. Hence the ANN can recognise nonlinear relations between the input variables and output variables, where a linear model might not.
This is a reason why we hypothesize that the ANN can be more useful for the prediction instead of an arbitrary linear model. However, it would be interesting if a linear model does exists which gives good results in combination with the attributes we applied in the hybrid model. We find there is a significant residual if the linear model ARIMA is applied and it is worth to improve this.

**Changes in manuscript**

We will write in the discussion of the revised manuscript a reason why the combination of the attributes and machine learning works well. Besides, we note in the discussion that a combination of attributes and a linear prediction model could be interesting.

*(iv) It would help to have a single final plot showing rms error vs prediction horizon as compared with the current methods.*

**Reply**

In the original manuscript we decided to only compare with the CFSv2 ensemble. We have thought of making a comparison with other conventional prediction methods such as in [1]. However, this requires that we know the rms error of the other prediction models for the same period or a subset of the period we have predictions for, since comparing the rmse obtained from predictions at different periods could be misleading.

**Changes in manuscript**

No changes will be made in the revised manuscript regarding this comment.

*1. P2, 1st line, not quite sure how to define "intuition and creative thinking", nor (more importantly) why this is relevant here.*

**Reply**

The Machine Learning method which competes with humans in the game GO is different.

**Changes in manuscript**

We will delete the whole sentence in the revised manuscript.

*2. P2, par lines 3-11, this seems a bit awkwardly worded. It isn't a binary choice between many layers and inputs and "simpler", but rather a continuum of choices with an inherent trade-off. Using more layers and input variables means you can rely more on the algorithm to figure out what matters at the expense of needing to train it on more training data, and the fewer variables/layers one uses the less training data might be required but the more that forces the user to make intelligent choices for input variables rather than relying on the algorithm to do so.*

**Reply**

We understand that the mentioned paragraph is confusing.

**Changes in manuscript**

We will rephrase the paragraph in the revised manuscript.

*3. The choices in Section 2.3 are not well motivated (that is, why are these the relevant choices to*

*feed into the ANN, and what else did you try?) This section could benefit from a couple of intro-
ductory sentences that describe the goal of the section, and the broad overview of the ideas of the
section.*

**Reply**

Section 2.3 includes the methods applied in the network analyses. It resulted in some variables showing interesting
properties of climate networks, but only one of them ($c_2$) is eventually applied in the ANN.

**Changes in manuscript**

In the revised manuscript all methods which are not applied in the prediction will be moved to the appendix. The
new section 2.3 only presents the method to calculate $c_2$ and it should be clear now from this section why it could
be useful for a prediction.

*4. Why is it adequate to have all of the memory embedded in the linear part of the model?*

**Reply**

To embed the memory only in the linear part of the model is a choice.
Two methods have been considered to include memory in the ANN. The first is the time delay neural network
(TDNN), where also lags of attributes are used as input variable. The second is a recurrent neural network (RNN),
where one allows loops in the neural network structure. We decided to stay with the feed-forward ANN, because
the other two types of neural networks would only complicate the hyperparameter tuning (i.e. for the TDNN one
has to decide which lags to implement and in the RNN the possible different structures increases), and embed all
histroy in the linear part of the prediction model.
In future research both TDNN and RNN could be interesting to apply, however we got interesting results with only
embedding the memory in the linear part.

**Changes in manuscript**

No changes will be made in the revised manuscript regarding this comment.

*5. For that matter, not entirely obvious to me, since you are using ML to predict the nonlinear
terms anyway, whether the ML can also predict the linear (but dynamic) part without any extra
effort, or for that matter the nonlinear and dynamic part. Did you try different things and conclude
you didn't have enough training data to converge, and kept simplifying, or did you just guess what
might work and then it did? I didn't go back and read Hibon and Evgeniou, but it would seem like the
question of how to simplify what the ML is actually learning is case dependent rather than absolute.
Some more motivation here is required (and at a minimum you should clarify what is meant by
"more stable" and provide a few more words of intuition as to why this reduces the risk of a bad
prediction.)*

**Reply**

We were looking for an easy method to implement the history in our prediction besides the feed-forward ANN,
which resulted in ARIMA as easiest and most straightforward method. Using only the feed-forward ANN did not
result in a good prediction.
'More stable' implies here that applying a combination of different types of prediction models, rather than only one
type of prediction model, decreases the variability of the prediction skill when both are applied to several arbitrary
time series.

**Changes in manuscript**

In the revised manuscript we will provide the motivation for the model choice. We also clarify what is meant by
'more stable' and why this reduces the risk of a bad prediction.

*6. Extra plus sign in eqn 13 and 14. Also, shouldn't the summation on the second term start*

*at d+1 (otherwise, the j=1 in the second term and the i=1 in the first term are identical, and you have a standard ARMA model rather than an ARIMA model). (Also, don't recall if you said why you were using ARIMA rather than ARMA?)*

**Reply**

Thank you for noticing this error. It is true that the differencing part is not incorporated well in this definition.

**Changes in manuscript**

We will use the definition similar to the definition in [2] in the revised manuscript, in order prevent any mistakes.

*7. P7, L19-20, why would including past El Nino and La Nina information reduce prediction skill?*

**Reply**

We hypothesize that the long-term memory, i.e. of previous La Niña and El Niño events, could contain information that is not relevant for the prediction of the coming year, because this information is not relevant anymore for the outcome in a chaotic system which is forced by high frequency noise.

**Changes in manuscript**

In the revised manuscript we will change the wording, such that the focus will be on the 'too long ago,' and not on the 'previous La Niña and El Niño events.'

*8. P8, L1, I'd have just thought the choice of lead time is like a choice of different variables, that there's nothing wrong with including the same variable at different times as part of the input.*

**Reply**

In this sentence we try to tell that at a specific lead time, one needs an optimal attribute set to optimize the prediction. This does not imply that an attribute cannot be used at several lead times.

**Changes in manuscript**

To prevent any misunderstanding the sentence will be rephrased to: 'Moreover, at every lead time an optimal attribute must be selected.'

*9. P8, L17, "generally" as in, "in this paper", or "generally" as in "in most research"?*

**Reply**

"Generally" as "in this paper" applies here.

**Changes in manuscript**

We will replace "Generally" by "in this paper" in the revised manuscript.

*10. Section 3.1, any reason why you only used 45 years of ZC output? Why not use a few thousand years of output? (I ran it for that long quite a long time ago, so I know it isn't a computational challenge to do.)*

**Reply**

We used only 45 years of data, because this comprises more then 10 ENSO cycles and this should be enough for the analyses we applied to the model. We recall that our main interest is to make predictions from the observational data, and in the observations we do not have much longer time series.'

**Changes in manuscript**

No changes will be made in the revised manuscript regarding this comment.

*11. Also, section 3.1, you might want to say up front a bit more about motivation -are you trying to learn from ZC which variables are best to use, or ultimately comparing predictive capability on ZC vs the real world, or get a good initial estimate of ANN weights from ZC so that you don't have to converge as much when you apply to the real world? These are all possible goals, but other than the second one, may be problematic if the physics in ZC doesn't match the real world physics (and while with their original parameter choices the equilibrium point in ZC is unstable with a chaotic self-sustained response, I think the general consensus now is that the real world isn't exhibiting chaos but rather stochastically forced response of a damped stable system). This is similar to the comment on Section 2.3; it would be helpful to have a few additional sentences that talk about where you're going with a section, why is it here, what are you hoping to learn, and what the structure of the section is. (I note subsequently that you never actually look at the predictability of CZ model, improvement thereof with ANN, and you also don't use the same variables in the real world analysis. . . can you be clear as to why this section is here and what you learned? Is it here just because you spent a lot of time on it and figure that should be documented somewhere, or is it essential to motivate the analysis of the real world?)*

**Reply**

It is true that the physics of the ZC model does not completely match the physics in the real world. However, we found a network variable in the ZC model which showed the same behaviour as in the observations (i.e. $c_2$).

**Changes in manuscript**

In the revised manuscript we will explain in the end of section 2.2 how the ZC model helped us to get to the finally applied attributes (as is explained in the beginning of this reply at comment (i)). Section 3.1 will change, since all network variables which are not applied in the prediction will be moved to the appendix, and it will be made made clear why the network variable that was used can be important for prediction.

*12. P10, L2, I think what you mean here is something like "when the ENSO index changes from increasing to decreasing (peak El Nino) or from decreasing to increasing (peak La Nina)"? (The wording is a bit unclear to me.) Similarly line 7, refer to the derivative of the ENSO index, rather than the derivative of ENSO. . . (to me, "ENSO" refers to the overall dynamic phenomenon, which isn't a thing that has a sign or a derivative).*

**Reply**

We understand the confusion.

**Changes in manuscript**

We will change the wording in the revised manuscript.

*13. Section 4, rather than just focusing on a few things like 2010 (which is cherry-picked as a year where the default scheme does badly), and a few prediction horizons, one thing that would help evaluate this method would be a single plot of rms prediction error versus time for the two methods (that is, for any month once you have sufficient past data, do the N-month prediction for every N up to a year or more using both methods, and then over this big set of month N predictions, what's the rms error?) This would also be a great way to compare your ARIMA alone with ARIMA + ANN.*

**Reply**

This figure was presented to show that the hybrid model can improve the other models drastically in a specific case. We agree that a figure showing the rmse at different lead time predictions could be nice to compare results.

However, this will require additional tuning at the different lead times. Besides, we will need the rmse of the other models in the same time interval at all these lead times, which we do not have.

The ARIMA prediction alone still had a very significant residual after the prediction.

**Changes in manuscript**

To compare the ARIMA only and ARIMA + ANN prediction, we will mention in the revised manuscript that this residual is very significant, which is a reason to add the ANN part in Sect. 4.

*14. P14, L11, what do you mean by "best-performing"? What metric? Does that mean that adding more neurons made it worse? Or do you just mean that adding more neurons didn't make it better?*

**Reply**

Here it means this ANN structure resulted in the lowest NRMSE from the ensemble of different ANN structures.

**Changes in manuscript**

This will be stated in the revised manuscript.

*15. P15, L4, why compare the two methods at different lags instead of the same lag?*

**Reply**

We compared two different lags here, because we only have the three month lead prediction instead of the four month lead prediction of the CFSv2 ensemble. In the hybrid model on the other hand, the attribute set resulted in a better result at the four month lead prediction compared to the three month lead hybrid model prediction, because the attribute set apparently contains better information four months ahead.

**Changes in manuscript**

No changes will be made in the revised manuscript regarding this comment.

*16. P15, L7, doesn't this contradict the abstract?*

**Reply**

We do not think this sentence contradicts the abstract.

**Changes in manuscript**

To be sure we are clear, however, we will change the wording of this sentence.

*17. P15, L14, I'm confused by this sentence -you do a better job at predicting things 1 year in advance than 6 months?*

**Reply**

That is true. This is possible because the ANN is trained at a specific lead time, say $n$ months ahead. The ANN hence gives a function from the attributes to the output $n$ months later. It is possible that the ANN trains better with the attributes at longer lead times. In this case, the attribute set also changed (i.e. the WWV is replacd by $c_2$), such that $c_2$ has this longer memory to improve the predictions longer ahead.

**Changes in manuscript**

No changes will be made in the revised manuscript regarding this comment.

*18. Also, I must have missed something; I thought you'd already picked the set of input variables, and now it sounds like you are only using a subset, and a different subset for each prediction*

*horizon. Overall, this sounds incredibly fragile. You do a lot of work to pick a few really good input variables, and any time you change the time horizon you might need to change those, and change the number of neurons. . . I thought the whole point of ANN was the ability to be lazy and let the algorithm do all the work for you!*

**Reply**

As explained in Sect. 3.2, we use cross-correlation and Wiener-Granger causality to determine the information of attributes at different lead times. Since the attribute set at the shorter lead times does not work well at larger lead times, we replace the WWV by $c_2$, which are physically related to each other.

In our method we cannot be just lazy. As explained in the introduction, we are not applying deep learning. In deep learning, where the ANNs are large in size, more attribute selection/reduction is done by the algorithm itself. The smaller the ANNs that are used, the more effort has to be done in the attribute selection. We apply this method instead of deep learning because the observational time series are relatively short. Deep learning is only known to work well if a lot of data is available.

**Changes in manuscript**

No changes will be made in the revised manuscript regarding this comment.

*19. P15, L15-16, again, I'm a bit confused. . . why do we need to maintain a whole ensemble of different ANN structures? This doesn't converge to something with enough neurons? Also, Figure 11, am I interpreting this right that you found a bunch of possible ANN structures that outperform the ones in Figure 9? (Sorry, I'm totally lost at this point so this might be off-base and simply imply some insufficient description.) Why not go back and redo Fig 9 with the better ANN structure? This entire section reads a bit as a collection of odds and ends of results rather than as a post-facto summary.*

**Reply**

The purpose of considering an ensemble of different ANN structures in Fig. 10 is that it shows the outcome is not sensitive to the specific ANN choice. This implies that the prediction shown in Fig. 9 was not a lucky shot, but more ANN structures converge to similar predictions.

In Fig. 11, we perform the cross-validation for the models of Fig. 9 for different training and test sets. This means we apply exactly the same attributes and ANN structures as in Fig. 9. We find that the models from Fig. 9 can perform better if different parts of the total time series are used as test and training set.

The section is meant to give the final prediction results, followed by a generalisation and validation of these results.

**Changes in manuscript**

In the revised manuscript, we will make sure the purpose of this section is made clear in the beginning of section 4. Besides, all comments/questions regarding this section will be addressed.

**References**

[1] Anthony G. Barnston, Michael K. Tippett, Michelle L. L'Heureux, Shuhua Li, and David G. Dewitt. Skill of real-time seasonal ENSO model predictions during 2002-11: Is our capability increasing? *Bull. Am. Meteorol. Soc.*, 93(5), 2012.

[2] Yi Shian Lee and Lee Ing Tong. Forecasting time series using a methodology based on autoregressive integrated moving average and genetic programming. *Knowledge-Based Syst.*, 24(1):66–72, 2011.